# Valorization of Peels of Eight Peach Varieties: GC–MS Profile, Free and Bound Phenolics and Corresponding Biological Activities

**DOI:** 10.3390/antiox12010205

**Published:** 2023-01-16

**Authors:** Dasha Mihaylova, Aneta Popova, Ivelina Desseva, Ivayla Dincheva, Yulian Tumbarski

**Affiliations:** 1Department of Biotechnology, University of Food Technologies, 4002 Plovdiv, Bulgaria; 2Department of Catering and Nutrition, University of Food Technologies, 4002 Plovdiv, Bulgaria; 3Department of Analytical Chemistry and Physical Chemistry, University of Food Technologies, 4002 Plovdiv, Bulgaria; 4Department of Agrobiotechnologies, AgroBioInstitute, Agricultural Academy, 1164 Sofia, Bulgaria; 5Department of Microbiology, University of Food Technologies, 4002 Plovdiv, Bulgaria

**Keywords:** peel, waste recovery, valorization, peach, free and bound phenolics, metabolites

## Abstract

Sustainability, becoming essential for food processing and technology, sets goals for the characterization of resources considered as food waste. In this work, information about the GC-MS metabolites of peach peels was provided as a tool that can shed more light on the studied biological activities. In addition, distribution patterns and contribution of the chemical profile and free and bound phenolic compounds as antioxidant, antimicrobial, and enzymatic clusters in peach peels of different varieties of Bulgarian origin were studied. The two applied techniques (alkaline and acid hydrolysis) for releasing the bound phenolics reveal that alkaline hydrolysis is a better extraction approach. Still, the results indicate the prevalence of the free phenolics in the studied peach peel varieties. Total phenolics of peach wastes were positively correlated with their antioxidant activity. The antioxidant activity results certainly defined the need of an individual interpretation for each variety, but the free phenolics fractions could be outlined with the strongest potential. The limited ability of the peels’ extracts to inhibit α-amylase and acetylcholinesterase, and the moderate antimicrobial activity, on the other hand, indicate that the potential of peach peels is still sufficient to seek ways to valorize this waste. Indeed, this new information about peach peels can be used to characterize peach fruits from different countries and/or different food processes, as well as to promote the use of this fruit waste in food preparation.

## 1. Introduction

Traditionally, plants are known to be abundant in secondary metabolites [1]. Several studies have designated the use of plants (including fruits and vegetables) rich in bioactive compounds in the management of non-communicable diseases [2,3]. Some of them are also produced in their fruits [4]. Knowing the distribution of these metabolites in different parts of the fruit is a piece of valuable information, especially in the light of a circular economy and waste recovery. That is why research is no longer limited to the edible part of the fruit, but also to the generated by-products of fruit processing [5,6]. Moreover, fruit processing generates a large amount of waste. In fact, according to the FAO, fruit and vegetables account for up to 45% of food waste, in general [7]. Recently, many articles emphasize fruit waste as a source of health-promoting biologically active substances that could be identified, extracted, and further used [8,9,10,11]. This is a sustainable approach targeting the global goal for “zero waste” in the environment.

Phenolic compounds are common in a number of natural raw materials—fruits, vegetables, cereals, herbs, etc. [12,13]. They draw scientific interest due to their biological activities, including anti-inflammatory, antibacterial, neuroprotective, anti-coronavirus, antidiabetic, etc., as well as their powerful antioxidant properties [13,14,15]. Polyphenol compounds are considered as the potential active compounds that inhibit the enzyme acetylcholinesterase, associated with dementia and Alzheimer’s Disease (AD) prevention [16]. In addition, fruit isoflavonoids are studied for their pancreatic lipase inhibition [17]. Phenolic acids and flavonoids are some of the most recognizable phenolic compounds, which can be seen in soluble free, soluble esterified, and insoluble forms in plants. Polyphenols in the bound forms are covalently linked to the polysaccharide and protein structural components of the cell wall. In a number of in vitro antioxidant analyses, bound phenols have shown significantly higher antioxidant activity than that of soluble phenols [18]. Moreover, bound phenols can endure in both stomach and small intestine conditions, and reach the colon intact, where they have been released and exhibit bioactivity [19]. 

Depending on the species, fruits and vegetables’ edible parts contain between 6.5% and 76.3% of the total phenols in bound form [18,19,20]. Of particular interest are fruit peels, which are believed to hold a substantial amount of non-extractable phenols [21,22]. The contribution of non-extractable polyphenols to the total polyphenol content of common fruit peels in different fruits varies from 23 to 82% [21,22,23]. That is why analyzing only the free phenolics may underestimate the total phenolic content and the related activities. Moreover, approximately 24% of the phenolic compounds in fruits are considered to be present in a bound form [20]. Recently, researchers have shown an increased interest in the valorization and recovery of food waste [24,25]. Data shows that food waste, as well as being a global sustainability issue, can also present a valuable research topic.

Fruit and vegetable peels are one of the most widely neglected sources of beneficial substances. Thus, the objective of the present study was to estimate the phytochemicals present in peach peels of peaches, flat peaches, and nectarines from different varieties, as well as to shed light about the amount of bound and free polyphenolic compounds and their contribution to antioxidant action and enzyme inhibitory potential. This will elucidate this underestimated fruit waste as a source of biologically active substances and beneficial actions. Resulting data will help evaluate differences and similarity between peaches, flat peaches, and nectarines. Results will also promote the possible utilization of peach peels in various industries, i.e., functional food production and cosmetology, among others.

## 2. Materials and Methods

### 2.1. Fruit Samples

Peels from the “Filina” (F), “Ufo 4” (U), “Gergana” (G), “Laskava” (L), “July Lady” (JL), “Flat Queen” (FQ), “Evmolpiya” (Evm), and “Morsiani 90” (M) varieties grown in the same plantation were the objects of analysis. “Filina,” “Laskava,” “July Lady,” and “Evmolpiya” are peaches; “Ufo 4” and “Flat Queen” are flat peaches with white flesh; “Gergana” and “Morsiani 90” are nectarines [26]. No bactericides were applied to plants during testing. The samples were collected in 2021 at eating ripeness (fruit growth has stopped, yellow-orange color appeared, and softening of tissue occurred) in the Fruit Growing Institute, Plovdiv, BG.

A vacuum freeze dryer (BK-FD12S, Biobase, Jinan, Shandong, China) aided in the sample preparation. Different peel samples were first freeze-dried, then, powdered and kept prior to extraction.

### 2.2. Gas Chromatographic–Mass Spectrometry Analysis (GC-MS) 

Lyophilized material (50.0 mg) from each sample was exposed to the following procedure: 500.0 µL methanol, 50.0 µL ribitol, and 50.0 µL n-nonadecanoic acid (internal standards in concentration 1 mg/mL to quantify polar and non-polar metabolites) were added, then the resulting mixture was heated using a thermo shaker TS-100 (Analytik Jena AG, Jena, Germany) for 30 min at 70 °C/300 rpm. A volume of 100.0 µL water and 300.0 µL chloroform were added after cooling down to room temperature, then the mixtures were centrifuged (Beckman Coulter, Brea, California, USA) for 5 min at 22 °C/13,000 rpm. The upper phase was intended to analyze the polar (amino and organic acids, carbohydrates) compounds, whereas the lower phase studied the non-polar (saturated and unsaturated fatty acids) metabolites. The two attained phases were vacuum-dried at 40 °C in a centrifugal vacuum concentrator (Labconco Centrivap, Hampton, NH, USA). 

In order to extract the saturated and non-saturated fatty acids, 1.0 mL 2% H_2_SO_4_ in methanol was added to the dried residue of fraction “non-polar metabolites,” then the mixture was heated for 1 h at 96 °C/300 rpm on a Thermo-Shaker TS-100. After cooling to the room temperature, the resulting solution was extracted with 3 × 10.0 mL n-hexane. Organic layers combined were vacuum-dried at 40 °C in a centrifugal vacuum concentrator (Labconco Centrivap).

Prior to analysis by GC-MS, samples were derivatized by the following two procedures. Firstly, a 300.0 µL solution of methoxyamine hydrochloride (20.0 mg/mL in pyridine) was added to the fraction “polar metabolites,” and the mixture was heated on thermo shaker for 1 h at 70 °C/300 rpm. After cooling, 100.0 µL N,O-Bis (trimethylsilyl) trifluoroacetamide (BSTFA) was added to the mixture, then heated on the thermo shaker for 40 min at 70 °C/300 rpm. Lastly, 1.0 µL from the solution was injected in the GC-MS. 

Secondly, 100.0 µL pyridine and 100.0 µL BSTFA were added to the fraction “non-polar metabolites” and then heated on the thermo shaker for 45 min at 70 °C/300 rpm. Then, 1.0 µL from the solution was injected in the GC-MS. 

GC-MS analysis was carried out using a gas chromatograph 7890A (Agilent) coupled to a mass selective detector 5975C (Agilent) and HP-5ms silica-fused capillary column coated with 0.25 µm film of poly (dimethylsiloxane) as the stationary phase (Agilent), 30 m × 0.25 mm (i.d.). The oven temperature program used was as follows: initial temperature 100 °C for 2 min, then 15 °C/min to 180 °C for 2 min, and after that, 5 °C/min to 300 °C for 10 min, run time 42 min. The flow rate of the carrier gas (Helium) was maintained at 1.2 mL/min. The injector and the transfer line temperature were kept at 250 °C; EI energy: 70 eV, mass range: 50 to 550 *m*/*z* at 1.0 s/decade. The temperature of the MS source was 230 °C. The injections were carried out in a split mode 10:1; the injection volume was 1 µL. 

Version 2.64 of the AMDIS software, (Automated Mass Spectral Deconvolution and Identification System, NIST, Gaithersburg, MD, USA) facilitated the comprehension of the obtained mass spectra and the recognition of the metabolites. Kovats retention index (RI) with reference compounds in the Golm Metabolome Database (http://csbdb.mpimp-golm.mpg.de/csbdb/gmd/gmd.html, accessed on 25 March 2022) and NIST’08 database (NIST Mass Spectral Database, PC-Version 5.0, 2008 from National Institute of Standards and Technology, Gaithersburg, MD, USA) were compared to the GC-MS spectra. The 2.64 AMDIS software verified the RIs of the compounds with a standard n-hydrocarbon calibration mixture (C_8_–C_36_, Restek, Teknokroma, Spain).

### 2.3. Free and Bound Phenolic Compounds Extraction

#### 2.3.1. Extraction of Free Phenolic Compounds

Free phenolic compounds of each variety’s peels were extracted in triplicate as follows: 0.5 g of sample was mixed with 10 mL 80% (80:16, *v*/*v*) ethanol, extracted at 50 °C for 30 min under ultrasound (UST 5.7150 Siel, Gabrovo, Bulgaria) and centrifuged at 10,000× g for 20 min. Phenolic extracts were filtered using filter paper (Whatman No. 1) and evaporated until near dryness (RV 10, Ika, Staufen, Germany). The final volume of the extracts was adjusted by adding 10 mL of 85% methanol (85:15, *v*/*v*) and stored at −20 °C until further analysis.

#### 2.3.2. Extraction of Bound Phenolic Compounds 

##### Alkaline Hydrolysis Method

Bound phenolic extracts were obtained according to the method described by Ding et al. [27] with modifications. The residues of the extraction process for free phenolics were subject to 18 h digestion with shaking at 30 °C, using 2 M sodium hydroxide (25 mL) and under a stream of nitrogen gas. The samples were acidified to pH 1.5–2.0 with 6 M hydrochloric acid and then extracted six times with 25 mL ethyl acetate. The upper layer was collected each time and combined. The ethyl acetate extracts were dried to complete dryness using a rotary evaporator at 38 °C (RV 10, Ika, Staufen, Germany). The dried bound extracts were reconstituted in 10 mL 85% HPLC grade methanol (85:15, *v*/*v*) and stored by avoiding light at −20 °C until analysis. 

##### Acid Hydrolysis Method

Bound phenolic compounds of each variety were extracted using the method reported previously [28]. The residues of the extraction process for free phenolics were treated with 25 mL of methanol/H_2_SO_4_ (90:10, *v*/*v*) at 70 °C for 24 h as the first hour was with sonication, and the resulting mixtures were neutralized with 10 M sodium hydroxide to pH 12.0 before being extracted six times with ethyl acetate. The supernatants were combined/merged and vacuum evaporated to dryness at 38 °C (RV 10, Ika, Staufen, Germany) before being reconstituted with 10 mL methanol/water (85:15, *v*/*v*) and stored by avoiding light at −20 °C until analysis.

### 2.4. Determination of Total Phenolic Contents (TPC)

A modified method of Kujala et al. [29] was used to analyze the TPC as described by Mihaylova et al. [30]. 

### 2.5. Determination of Total Flavonoid Content (TFC)

The total flavonoid content was assessed following the description of Kivrak et al. [31]. Results are expressed as μg quercetin equivalents (QE)/g dw, as quercetin (QE) was used as a standard.

### 2.6. Determination of Total Monomeric Anthocyanins (TMA)

The TMA content was defined using the pH differential method [32]. Results are expressed as µg cyanidin-3-glucoside (C3GE)/g dw.

### 2.7. Evaluation of Antioxidant Activities of Phenolic (Free and Bound) Fractions

#### 2.7.1. DPPH^•^ Radical Scavenging Assay

The slightly modified method of Brand-Williams et al. [33], as described by Mihaylova et al. [30], aided in the identification of the capability of the extract’s to donate an electron and scavenge 2,2-diphenil-1-picrylhydrazyl (DPPH) radicals. The antioxidant activity is presented as a function of the concentration of Trolox with equivalent antioxidant activity expressed as μM TE/g dw.

#### 2.7.2. ABTS^•+^ Radical Scavenging Assay

The method of Re et al. [34] was used to estimate the extracts’ ABTS^•+^ radical scavenging activity. The results are expressed as μM TE/g dw with Trolox as standard.

#### 2.7.3. Ferric-Reducing Antioxidant Power (FRAP) Assay

The procedure of Benzie and Strain [35], with slight modification as described by Mihaylova et al. [30], was carried out in the FRAP assay, recording the absorbance at 593 nm, and expressing the results as μM TE/g dw with Trolox as standard.

#### 2.7.4. Cupric Ion-Reducing Antioxidant Capacity (CUPRAC) Assay

The CUPRAC assay followed the procedure of Apak et al. [36]. Results are expressed as μM TE/g dw with Trolox as standard.

### 2.8. Enzyme-Inhibitory Activities

The Sigma Aldrich method [37] specified by Mihaylova et al. [24] was used to carry out the α-Amylase (AM)-Inhibitory Assay. An α-Glucosidase (AG)-Inhibitory Assay was completed as described in the paper by Mihaylova et al. [26]. The in vitro pancreatic-lipase-inhibitory activity was assessed as described by Saifuddin et al. [38] and Dobrev et al. [39], with modifications explained in previous research [26]. The experimental conditions of the in vitro AChE-inhibitory assay were built on the method defined by Lobbens et al. [40], with modifications available by Mihaylova et al. [26]. All results are expressed as the concentration of extract (IC_50_) in mg/mL that inhibited 50% of the respected enzyme (α-amylase, α-glucosidase, lipase, and acetylcholinesterase).

### 2.9. Antimicrobial Activity

#### 2.9.1. Test Microorganisms

Four Gram-positive bacteria (*Bacillus subtilis* ATCC 6633, *Staphylococcus aureus* ATCC 25923, *Listeria monocytogenes* NBIMCC 8632, *Enterococcus faecalis* ATCC 19433), four Gram-negative bacteria (*Salmonella enteritidis* ATCC 13076, *Escherichia coli* ATCC 8739, *Proteus vulgaris* ATCC 6380, *Pseudomonas aeruginosa* ATCC 9027), two yeasts (*Candida albicans* NBIMCC 74, *Saccharomyces cerevisiae* ATCC 9763) and six fungi (*Aspergillus niger* ATCC 1015, *Aspergillus flavus*, *Penicillium* sp., *Rhizopus* sp., *Mucor* sp.-plant isolates, *Fusarium moniliforme* ATCC 38932) from the collection of the Department of Microbiology at the University of Food Technologies, Plovdiv, Bulgaria, were selected for the antimicrobial activity test. 

#### 2.9.2. Culture Media

Luria–Bertani agar medium supplemented with glucose (LBG) was prepared as prescribed by the manufacturer (Laboratorios Conda S.A.): 50 g of LBG-solid substance mixture was dissolved in 1 L of deionized water. The final pH was adjusted to 7.5, then the medium was autoclaved at 121 °C/20 min.

Malt extract agar (MEA) was prepared as suggested by manufacturer (HiMedia^®^, Thane, India): 50 g of the MEA-solid substance mixture was dissolved in 1 L of deionized water. pH was corrected to 5.4 ± 0.2, and then the medium was autoclaved at 115 °C/10 min.

#### 2.9.3. Antimicrobial Activity Assay

The agar well diffusion method [41] was implemented in the antimicrobial activity determination. The test bacteria *B. subtilis* was cultured on LBG agar at 30 °C; *S. aureus*, *L. monocytogenes*, *E. faecalis*, *S. enteritidis*, *E. coli*, *P. vulgaris*, and *P. aeruginosa* were cultured on LBG agar at 37 °C for 24 h. The yeast *C. albicans* was cultured on MEA at 37 °C, while *S. cerevisiae* was cultured at 30 °C for 24 h. The fungi *A. niger*, *A. flavus*, *Penicillium* sp., *Rhizopus* sp., *Mucor* sp., and *F. moniliforme* were grown on MEA at 30 °C for 7 days or until sporulation. 

A small amount of biomass in 5 mL of sterile 0.5% NaCl was homogenized to prepare the inocula of test bacteria/yeasts, while 5 mL of sterile 0.5% NaCl was placed into the tubes for the test fungi. After stirring by vortex V-1 plus (Biosan, Riga, Latvia), they were filtered and transferred in other tubes prior to usage. A bacterial counting chamber Thoma (Poly-Optik, Germany) established the number of viable cells and fungal spores. Their final concentrations were adjusted to 10^8^ cfu/mL for bacterial/yeast cells and 10^5^ cfu/mL for fungal spores, and then inoculated in agar media that was preliminarily melted and tempered at 45–48 °C. The inoculated media were subsequently transferred in a quantity of 18 mL, in sterile Petri plates (d = 90 mm) (Gosselin™), and allowed to harden. After that, six wells (d = 6 mm) per plate were cut, and triplicates of 60 μL of the extracts were pipetted into the agar wells. The Petri plates were incubated at identical conditions.

The inhibition zones’ diameters around the wells were measured twice on the 24th and 48th hours of incubation to establish antimicrobial activity. Test microorganisms with inhibition zones ≥ 18 mm were regarded as sensitive; those with zones ranging from 12 to 18 mm were considered moderately sensitive; and those with zones ≤ 12 mm were considered resistant. 

### 2.10. Statistical Analyses

Each sample was triplicated and the results are expressed as the mean ± SD. The impact of the peach variety and extraction type on the TPC, TFC, TMA, and AOA was evaluated using a two-factor variance analysis [42]. The Tukey–Kramer post hoc test (α = 0.05) [42] was used to statistically compare the data. The web-based MetaboAnalyst platform (www.metaboanalyst.ca, accessed on 27 June 2022) [43] was used to conduct the PCA and HCA of GC-MS data, as previously described by Mihaylova et al. [26]. 

## 3. Results and Discussion 

### 3.1. GC-MS Volatile Profile Characterization of Analyzed Peach Peels

Due to the beneficial properties of metabolites, the interest in metabolite profiling is constantly growing. Presenting information about the metabolite profile of different parts of the fruit is a piece of useful knowledge, especially in the light of waste recovery and resource scarcity. That is why research is no longer limited to the plant’s edible part, but also includes the generated by-products of fruit processing, such as peels, stones, and pressed pulp, among others. In the view of the abovementioned, a semi-quantification GC-MS profile aids in the characterization of the peels of eight peach varieties. The identified metabolites (Table 1) in the current study are divided into five groups: sugars and sugar-alcohols, organic acids, amino acids, phenolic acids, and fatty acids. It is important to highlight the availability of potentially active substances in the peels, as they are often regarded as food waste. 

Similar to the whole fruit [26], the peach peel is most abundant in sugars. Organic acids being intermediates in the degradation pathways of amino acids, fats, and carbohydrates, also affect properties such as the fruit’s color, flavor, and aroma [44]. The organic acids content in the studied peaches, flat peaches, and nectarines is relatively similar. The documented contents of organic acids confirm that the skin is flavor-contributing and might influence the overall consumer’s acceptance by referencing the color of the skin in terms of visual ripeness. Considering the amino acids content, it is insufficient for human daily needs, but it is visible that early ripening varieties have peels richer in amino acids compared to late ripening varieties. 

Shikimic acid, one of the major organic acids in all samples, has been recognized for its neuraminidase inhibition potential [45]. Shikimic acid has also presented its anti-inflammatory effect [46]. The predominant fatty acids in the studied peels are the saturated palmitic, stearic, and behenic acids, as well as the unsaturated linoleic, oleic, and arachidic acids. Some researchers [47] point out that oleic and linoleic acids possess powerful inhibitory effects on the α-glucosidase activity, but they are also competitive inhibitors, and their interactions with α-glucosidase showed a character of static quenching, which directs them to bind to α-glucosidase to form a complex. Recent research [48] highlights palmitic acid as a strong α-amylase inhibitor. Fatty acids also have high potency in the therapy of Alzheimer’s disease due to their inhibition of cholinesterases (AChe and BChe) [49]. Chlorogenic acid is a main phenolic compound frequently present in plants [50]. Expectedly, chlorogenic acid is a major compound identified in the extracts. It is a result from the esterification of caffeic with quinic acid. The production of chlorogenic acid reduces the ability of caffeic and quinic acids to inhibit α-amylase and α-glucosidase [51]. 

### 3.2. Total Phenolic, Flavonoid and Total Monomeric Anthocyanins Contents of Free and Bound Insoluble Fractions

Phenolic compound are responsible for both the desirable and undesirable qualities of the peach fruit [52]. The presented GC-MS profile of the peels revealed the free phenolic acids (Table 1). Although they are in relatively small amounts, it is important the clear out their distribution in such parts of the fruit that are often left unconsumed. Phenolic compounds are one of the main classes contributing to the biological activity of plant matrices and fruits, in particular [53]. Traditional solvent extraction usually omits high quantities of bound phenolics that play an essential role in human health benefits. Due to their beneficial effects, the use of the full spectrum of polyphenolic potential has attracted considerable attention, and it seems reasonable to apply different recovery techniques. The peach fruit, itself, is reported as an excellent supply for phenolic components [54]. Peach peels, on the other hand, are yet to be thoroughly characterized. Bearing in mind the abovementioned, in targeting to reveal the biological potential of the peach peels, several phenolic compounds profile was assessed. 

Based on existing reports [55,56] about the richness of stone fruits in phenolic compounds, the current study focused on the evaluation of their distribution in both soluble and non-soluble forms. Various techniques, including alkaline, acid, enzymatic, and ultrasound-assisted hydrolyses, can be applied in order to release the bound insoluble phenolic fractions from the cell wall [18].

A two-way ANOVA aided in the evaluation of the peach variety and type of extraction on the TPC, TFC, and TMA. The interpretation of the results showed that the single effect of both factors and their combination was influential (*p* < 0.05) to the TPC, TFC, and TMA. The highest TPC was found in soluble phenolics extracts, showing the relationship of the total phenolic content in peach fruits with the extractable free polyphenols (Figure 1). The total TPC of the samples varied between 15.56 and 20.49 mgGAE/g dw, accounting mainly of extractable polyphenols—from 42 to 76%. The TPC of the free soluble polyphenols was in the range of 6.82 ± 0.13 to 13.12 ± 0.09 mgGAE/g dw. Previous research also points out that free phenolic compounds are predominant in plant-based extracts [57,58]. The alkaline hydrolyzed non-extractable polyphenols account for 16 to 31%. Other authors reported alkaline treatment, as such, of low yield [6]. Depending on the fruit variety, acid and alkaline hydrolyzed polyphenols contribute in different manners to the fruit’s total TPC. Furthermore, the total phenolic content of the extractable polyphenols was statistically superior to that of non-extractable polyphenols within the same variety, valid for all eight. The samples with the highest TPC were the free fractions of the “Flat Queen” and “Laskava” varieties, and the lowest was the “Evmolpiya” variety. In terms of total TPC, the highest values were established in the “Flat Queen” variety, followed by “Laskava,” confirming the contribution of the free soluble polyphenols to the total TPC. The contribution of bound phenolics amounted to a range of 7 to 31% of the total phenolics. 

The current results suggest that the alkaline hydrolysis method was more effective compared to the acid one, and efficiently liberated bound phenolic compounds (Figure 1), which is comparable to other research targeting by-products, i.e., fruit peels [59,60]. Alkaline hydrolysis successfully breaks the ether and ester bonds which link phenolic compounds to the cell wall, and are commonly spread in fruit peels [18]. This might explain the established trend in the current study.

Flavonoids are the largest group of polyphenols. In respect to the total flavonoid content, the free phenolics extracts are with the predominant content in most of the varieties, ranging from 164.14 ± 4.72 (“Ufo 4”) to 515.83 ± 30.59 μgQE/g dw (“Flat Queen”). The content of flavonoids was higher when alkaline hydrolysis was applied (Figure 1) for “Ufo 4” and “July Lady” compared to the soluble phenolics extracts. The contribution of acidic hydrolyzed flavonoids could be neglected as below the limit of detection for all the varieties. The TFC of the peach peels varied from 380.58 to 999.38 μgQE/g dw. The alkaline hydrolyzed fraction displayed a TFC of 0 and 76.93% from the total TFC of the samples. The established results follow the same trend as for the total phenolic content. Other researchers also acknowledge the fact that flavonoids in free form are predominant in plant-based matrices [61].

The total monomeric anthocyanins content was mainly due to the free extractable polyphenolics fraction, and the total content was in the range from 327.84 to 1246.77 µgCya-3-glu/g dw (“Gergana”). The distribution between soluble and insoluble phenolics (Figure 1) showed no or limited contribution of the acid hydrolyzed phenolics, and relatively low input of the alkaline hydrolyzed ones, which is not surprising due to the degradation of the anthocyanin content with the increased temperature and pH value [62].

About 1%, 7%, 31%, 17%, 40%, 49%, 11%, and 0% of TAC were present in bound form in the peels of the investigated varieties. The uneven distribution of TAC among soluble and insoluble phenolics fractions confirms the need for personalized/individual evaluation of the potential of peel waste as a source of biologically active substances. 

In brief, most of the analyses are synchronous regarding the predominance of the potential of free phenolic extracts, revealing the effectiveness of commonly applied extraction techniques. In the current study, alkaline hydrolysis resulted in more bound phenolics, flavonoids, and monomeric anthocyanins compared to acid hydrolysis. Earlier studies have established that alkaline hydrolysis is a more comprehensive bound phenolics extraction technique than acid hydrolysis [60,63,64]. This might be attributable to the fact that alkaline hydrolysis has the ability to split the ester bonds between phenolic acid and polysaccharide, and decrease phenolic acids losses [65,66]. However, acid hydrolysis mainly breaks glycosidic bonds. Contrary to the abovementioned, far more bound phenolic compounds were released by acid hydrolysis from litchi pulp extracts [67] and apple and peach [68], which obviously indicates the need for optimal extraction conditions in each particular study. The variation in the food matrices, as well the difference in the bond types of the bound phenolics, should have an effect on the extraction process. Additionally, Verma et al. [69] stated that elevated-temperature acid hydrolysis resulted in the loss of some phenolics. This may reason the better efficacy of the alkaline hydrolysis in the release of bound phenolics from the evaluated peach peels compared to acidic hydrolysis.

### 3.3. Antioxidant Activity (AOA)

The potential to recover the antioxidant activity of the peach peels is reported for locally grown peach varieties [70,71]. Moreover, authors revealed peach peels with higher antioxidant activity in comparison to the pulp, suggesting that the fruit peel is a nutrient carrier [70]. 

In the current study, the peach variety, the type of extraction, and their combination had impact (*p* < 0.05) on the AOA (DPPH, ABTS, FRAP, and CUPRAC assays) as studied by two-way ANOVA. The assessed in vitro antioxidant potential of the investigated peach peels is the strongest with regard to the free phenolics fractions (Figure 2). According to the DPPH and FRAP assays, the most potential extract was the free phenolics one, and the one of the late-season “Morsiani 90” variety, in particular. The free phenolic fraction of “Morsiani 90” was the most active one according to the CUPRAC assay as well, although several alkaline extracts had good prospective (Figure 2D). The antioxidant potential towards the DPPH free radical was in the range of 1.94 ± 0.04 to 43.43 ± 0.28 µMTE/g dw (Figure 2A). The FRAP assay showed values from 5.14 ± 0.18 to 123.08 ± 4.0 µMTE/g dw (Figure 2C). According the CUPRAC assay, the results varied from 23.13 ± 0.84 to 122.97 ± 4.71 µMTE/g dw (Figure 2D). When the three abovementioned in vitro assays are applied, the free phenolics extracts show the greatest potential, significantly different from the bound phenolic fractions. The ABTS assay, for example, was revealed as most active for the early season varieties “Gergana,” “Filina,” and “Ufo 4” (free phenolics fractions).

The results for the bound phenolic compounds released by acid hydrolysis were substantially lower than those obtained by alkaline hydrolysis for most of the varieties. The discrepancy in the antioxidant activity results confirms the need for more than one assay to be applied, bearing in mind the different aspects of the antioxidant activity mechanism and contributing compounds, in particular.

While retrieving the bound phenolic fractions, the conducted alkaline and acid hydrolyses resulted in less antioxidant activity in most of the peach peel extracts. Furthermore, some authors correlate the total phenolic content and antioxidant potential decrease with ripening. Nonetheless, more than 50% of the total antioxidant activity is contributed by the extractable polyphenols [54]. However, the assessed activity is an important contribution to the general antioxidant potential of the whole peach fruit, and the peels in particular. In general, the alkaline and acid hydrolyzed fractions showed moderate activity, and no clear trend could be pointed out. This certainly outlined the need for individual interpretation of the results for each particular variety. 

Tang et al. [59] reported the better potential of the bound phenolics extracts of the pitahaya peel when the alkaline hydrolysis method was applied. More specifically, the authors established good correlation of the antioxidant activity and the highest phenolic content achieved. Furthermore, the paper revealed that hydrolysis methods had a significant effect on the release of phenolics, in preference of the alkaline method.

The present study is validating the fruit wastes potential and fruit peels, in particular. Therefore, the question set to peel or not to peel [72] seems to have an answer. Peach peels are worth researching and using, as the latter contribute to a more beneficial absorption of compounds when the unpeeled fruit is consumed.

### 3.4. Inhibitory Potential towards α-Glucosidase, α-Amylase, Lipase, and Acetylcholinesterase of Analyzed Prunus Persica Peels

Based on the results presented in Table 1 (GC-MS profile of the peels), indicating a possible inhibitory potential, extracts from the peels were analyzed for inhibitory activity against α-glucosidase, lipase, α-amylase, and acetylcholinesterase (Table 2). The results are expressed as the concentration in mg/mL that inhibits 50% of the corresponding enzyme activity. The current finding may be explained by the specific metabolite profile of the peels (Table 1). For example, the fatty acids content, which is relatively high in the studied peel extracts, may be the reason for the inhibitory potential towards acetylcholinesterase. 

The samples that are marked with “-” are not active, or it is impossible to calculate IC_50_. None of the extracts were able to inhibit the action of lipase. The action of alpha-glucosidase was suppressed by most of the samples. All extracts of “Ufo 4” and “Laskava” peels were active, resulting in IC_50_ concentration in the range of 5.9 to 39.7 mg/mL. The alkaline hydrolysis of bound phenolics of the “Morsiani 90” sample seems to possess the best inhibitory activity toward alpha-glucosidase—2.6 mg/mL. The alkaline hydrolysates of “Filina” and “July Lady” samples were both able to inhibit α-amylase, at IC_50_—20.55 ± 0.51 and 17.38 ± 0.11, respectively, and AChE at IC_50_—31.1 ± 0.22 and 17.8 ± 0.52, respectively.

### 3.5. Antimicrobial Activity of Peach Peel Extracts

It has been proposed that phenolic compounds (i.e., flavonoids and phenolic acids) can exhibit antimicrobial properties [73]. Thus, the antimicrobial activity of the peach peel extracts was evaluated (Table 3) as an indicator of the biological potential of the peach peels. No particularly high inhibition was detected to calculate the minimal inhibitory concentration (MIC). However, the antibacterial potential is valuable for fruits in order to recover injuries and/or to have prolonged shelf life [74]. In respect of both Gram-positive and Gram-negative bacteria and yeasts, the free phenolics fraction showed no activity. The inhibitory effect of the bound phenolics was more pronounced against *Bacillus subtilis* ATCC 6633, *Listeria monocytogenes* NBIMCC 8632, *Pseudomonas aeruginosa* ATCC 9027, and *Saccharomyces cerevisiae* ATCC 9763. Other authors have reported the antibacterial activity of *Prunus persica* varieties to be more sensitive against *Staphylococcus aureus* and *Listeria monocytogenes* [75]. This is consistent with the current findings, as well as their relation to the polyphenolic content and their antioxidant activity. Koyu et al. [76] also reached a minimum inhibitory activity for *Prunus* leaves against *Escherichia coli*, *Staphylococcus aureus*, *Staphylococcus epidermidis*, *Enterococcus faecalis*, *Enterococcus faecium*, and *Candida albicans*. In line with the reports of Mocanu et al. [77], the better reaction towards *B. subtilis* may be a result of elevated flavonoid concentrations. Molecular weight, polarity, and side groups command the specific inhibitory effect of each phenolic compound [78]. Phenolic compounds, such as acids (ferulic acid, p-coumaric acid, among others), alcohols (guaiacol, catechol, vanillyl alcohol), and aldehydes (vanillin, syringaldehyde), are regarded as the most potent inhibitors of microbial growth [79]. Ferulic and p-coumaric acids are the second and third most abundant in the studied samples after the chlorogenic acid (Table 3). Organic acids may be responsible for their activity against Gram-positive and Gram-negative bacteria, due not only to their quantity and diverse biochemical nature, but also their ability to lower pH [80].

Regarding the antifungal activity of the peach extracts, results show limited activity (Table 3). For instance, the bound phenolics fractions of all varieties possess better activity compared to the free phenolics fractions. The peach peels show better activity toward *Rhizopus* sp. and *Fusarium moniliforme*. 

None of the samples inhibited the growth of the fungi *Mucor* sp., and as regarding *A*. *flavus*, only bound phenolic extracts under alkali conditions exhibit activity (inhibition zones of 8 mm). Like in other reports [81], the current findings may suggest a link between the antimicrobial activity and the fatty acid and flavonoid content of the studied extracts.

### 3.6. Correlation between Phenolic Compounds Content and Antioxidant Activity

Several components are contributing to the antioxidant activity, usually such of phenolic nature. The carried correlation analysis is based on the Pearson correlation coefficient, also referred to as Pearson’s r, in order to express the strength and direction of the linear relationship of correlation. Total phenolics compounds content was positively and significantly correlated to all of the antioxidant assays (r = 0.5059–0.8856, *p* ≤ 0.01). The positive relationship between total phenolics content and antioxidant activity was also previously stated [59,82]. No significant correlation between TPC and TFC was observed (r = 0.3249, *p* > 0.05). No significant correlation was established between TFC and TMA, or between TFC and ABTS either, pointing out that the contribution of the TFC is relatively low to the established activities. Previous reports also [59] stated no significant correlation between TFC and the antioxidant activities in fruit peel extracts. The relatively low content of TFC in the free and bound fractions is possibly contributing to the weak or non-existent correlation between TFC and the other assays.

Among the antioxidant activity assays, the ABTS assay has a moderate correlation, significant at *p* ≤ 0.05, to the FRAP and CUPRAC. The ABTS was also significantly correlated to another antiradical assay (DPPH assay) at *p* ≤ 0.01. This pointed to the ABTS assay as a not very appropriate method by which to evaluate the potential of the particular free and bound phenolics in peach peels. The strongest positive correlation was observed between the DPPH and FRAP assays, and between the TPC and DPPH (Table 4). 

### 3.7. Principal Component Analysis (PCA) and Hierarchical Cluster Analysis (HCA) of GC-MS and Phenolic Compound and AOA Assays Data

In order to verify the sample differences or resemblances, principal component analysis (PCA) and hierarchical cluster analysis (HCA) of the volatile compounds identified were utilized. According to the PCA plot (Figure 3), the first two principal components PC1 (33.1%) and PC2 (18.6%) summed up 51.7% of the total variance of all identified volatile compounds in the analyzed peach peels. When taking into account the total flavonoid content, total phenolic content, total monomeric anthocyanins, and the antioxidant assays applied, the PCA plot reveals 50% of the total variance in the analyzed samples, namely 27.5% for PC1 and 22.5% for PC2.

High positive load scores in PC1 (Figure 3), which distinguish “Laskava” from the other studied peels, are shown by quinic acid, oleic acid, and fructose isomer 2. The high negative scores in PC1 clearly differentiate the “Filina” peels from the others. The “Gergana” peels stood out from the rest by the high negative scores of a number of amino acids in PC2. 

Figure 4A,B reveal the PCA score plots of TPC, TMA, TFC, and AOA assays of peach (*Prunus persica* L.) peels from the studied samples. The content of total monomeric anthocyanins, as well as the FRAP antioxidant assay, can be characterized as important for the varieties “Laskava” and “July Lady,” due to their high values. The total flavonoid content, on the other hand, is less dependent for the “Morsiani 90” peels. The ABTS values are not defining for the peels of the “Flat Queen” variety. 

The peels of the two nectarine varieties (“Gergana” and “Morsiani 90”) were grouped in the same cluster, due to their phytochemical similarity, while “July Lady” and “Laskava” were clustered in another. The results from the HCA show significant differences from the ones established for the whole fruit of the same varieties [83]. A clear distinction of nectarines compared to peach and flat peach peels is shown in the current results (Figure 5A). When taking into account the total flavonoid content, total phenolic content, total monomeric anthocyanins, and the antioxidant assays applied, the heatmap (Figure 5B) shows that peaches from the same ripening period are clustered together. 

The statistically independent variables (assays or compounds) can be drawn from the heatmaps (Figure 4 and Figure 5). Peels from each variety are characterized with different linear relationships. Considering the clade arrangement in the Figures, it can be assumed that phenolic compounds and antioxidant assays lead to more distinct differences between the established clades.

## 4. Conclusions

This study represents new information concerning the phytochemical peculiarities of peels from four native Bulgarian peach varieties and four introduced in the geographical region of the Thracian valley. Information about the GC-MS metabolites can shed more light on the studied biological activities. The current findings are one of the few reports on the topic of free and bound phenolics in several peach peel varieties, including flat peaches, peaches, and nectarines. The results are consistent with other existing literature stating that alkaline hydrolysis is a better extraction approach for distributing bound phenolics. The results also indicate the prevalence of the free phenolics in the studied peach peel varieties. The present findings confirm, yet again, the well-known facts about the health-promoting properties of polyphenols, and the fact that fruit by-products can provide potential accessible sources of antioxidants for direct consumption. Furthermore, peach peels could be considered useful natural sources of bioactive compounds with prospective activities. In any case, they are worth contemplating for waste recovery. This study can be seen as a stepping stone in the context of functional foods enriched with natural extracts obtained through effective extraction.

## Figures and Tables

**Figure 1 antioxidants-12-00205-f001:**
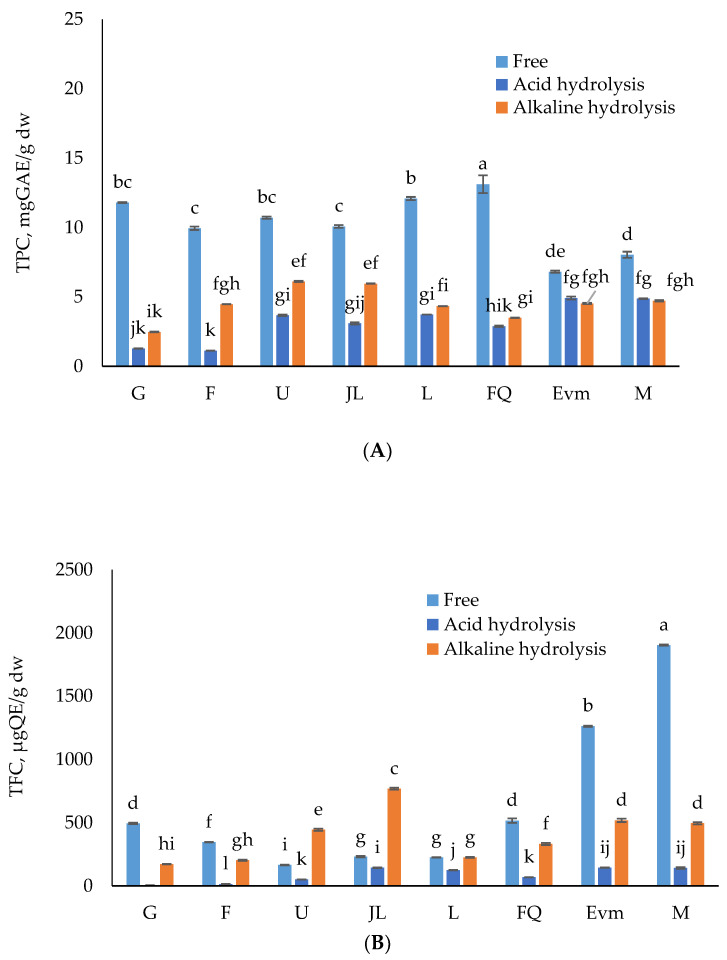
Free and bound (insoluble) phenolics content distribution of eight peach varieties’ peel extracts—(**A**) Total phenolic content (TPC, mgGAE/g dw), (**B**) Total flavonoids (TFC, μgQE/g dw) and (**C**) Total monomeric anthocyanins (TMA, µg cyanidin-3-glucoside (C3GE)/g dw). G—“Gergana”, F—“Filina”, U—“Ufo 4”, JL—“July Lady”, L—“Laskava”, FQ—“Flat queen”, Evm—“Evmolpiya”, M—“Morsiani 90”. Different letters (a–l) within chart columns indicate significant differences (*p* < 0.05) between treatments as analyzed by two-way ANOVA and the Tukey test (n = 3 per treatment group).

**Figure 2 antioxidants-12-00205-f002:**
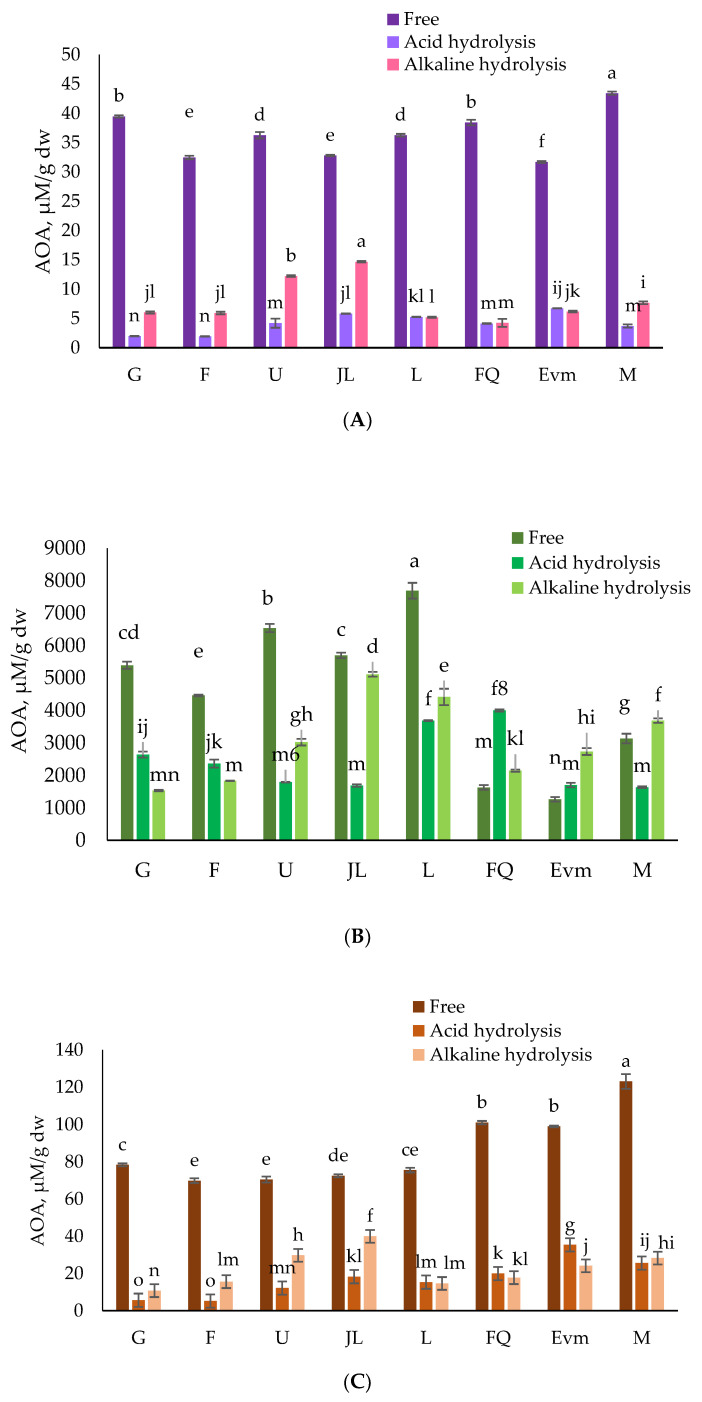
Antioxidant activity of free and bound phenolics in eight peach varieties’ peel extracts (µMTE/g dw) by (**A**) DPPH, (**B**) ABTS, (**C**) FRAP and (**D**) CUPRAC assays. Different letters (a–n) within chart columns indicate significant differences (*p* < 0.05) between treatments as analyzed by two-way ANOVA and the Tukey test (n = 3). G—“Gergana”, F—“Filina”, U—“Ufo 4”, JL—“July Lady”, L—“Laskava”, FQ—“Flat queen”, Evm—“Evmolpiya”, M—“Morsiani 90”.

**Figure 3 antioxidants-12-00205-f003:**
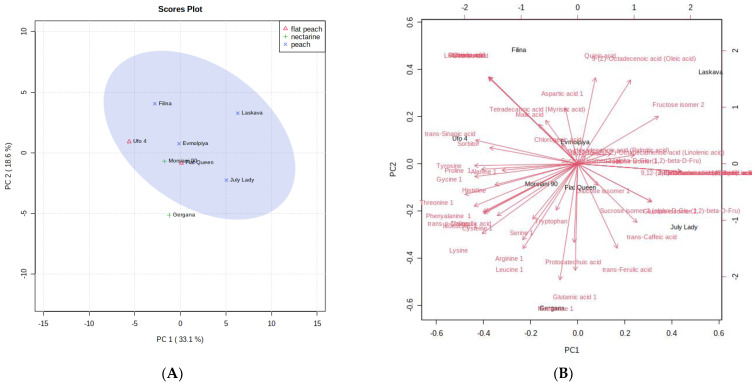
Principal component score plot (**A**) and eigenvector load values (**B**) of GC-MS data of volatile compounds of peach (*Prunus persica* L.) peels for the eight peach peel varieties.

**Figure 4 antioxidants-12-00205-f004:**
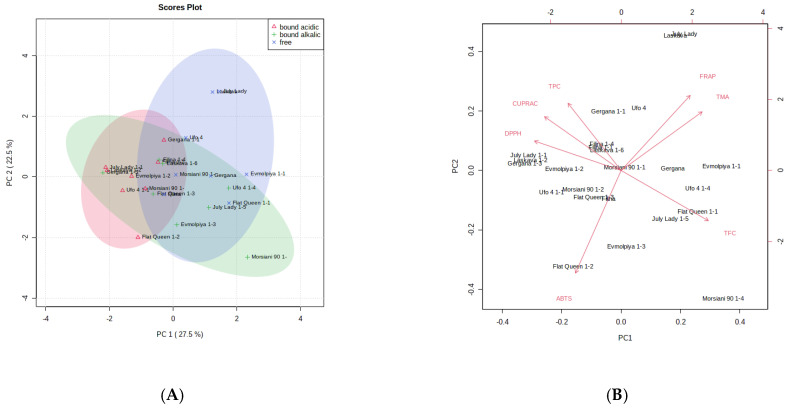
Principal component score plot (**A**) and eigenvector load values (**B**) of TPC, TMA, TFC, and AOA assays of peach (*Prunus persica* L.) peels for the eight peach peel varieties.

**Figure 5 antioxidants-12-00205-f005:**
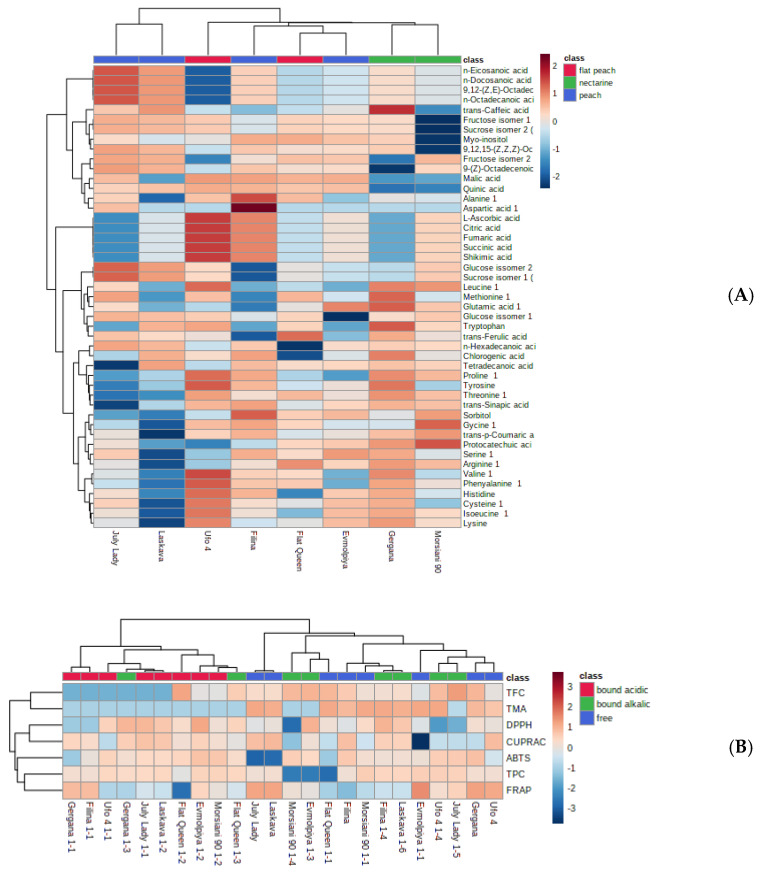
Heatmap of the clustering result of peach peels from eight varieties. (**A**) GC-MS data of volatile compounds and (**B**) TPC, TMA, TFC, and AOA assays. Values were normalized by log_10_ transformation.

**Table 1 antioxidants-12-00205-t001:** Semi-quantification of polar metabolites (mg/g dw) of eight peach varieties’ peel extracts.

	RI	Polar Metabolites	F	G	U	L	JL	FQ	Evm	M
**Amino acids**										
1	1097	Alanine	0.705	0.104	0.209	0.171	0.011	0.228	0.041	0.086
2	1208	** Valine **	0.142	0.307	0.549	0.079	0.014	0.123	0.027	0.052
3	1266	** Leucine **	ND	0.052	0.082	0.007	ND	0.010	ND	0.035
4	1285	** Isoleucine **	0.119	0.265	0.435	0.117	0.009	0.034	0.207	0.143
5	1293	Proline	0.161	0.297	0.525	ND	0.008	0.080	ND	0.115
6	1299	Gycine	0.224	0.079	0.177	0.059	ND	0.137	0.078	0.391
7	1351	Serine	0.591	0.618	1.240	0.438	0.028	0.359	0.711	0.239
8	1376	** Threonine **	0.201	0.932	0.777	0.600	0.011	0.477	0.178	0.417
9	1508	Aspartic acid	1.141	ND	ND	0.068	ND	ND	ND	ND
10	1515	** Methionine **	0.050	0.401	0.124	0.191	0.007	0.142	0.035	0.036
11	1550	Cysteine	0.007	0.014	0.024	0.009	ND	0.005	0.011	0.020
12	1609	Glutamic acid	0.280	0.522	0.860	0.175	0.051	0.137	0.389	0.217
13	1635	** Phenyalanine **	0.215	0.283	0.467	0.106	0.033	0.205	0.058	0.151
14	1833	Arginine	0.171	0.376	0.710	0.142	0.016	0.416	0.211	0.273
15	1910	** Lysine **	0.184	0.857	1.072	0.243	0.015	0.247	0.563	0.352
16	1930	Tyrosine	0.156	0.438	0.669	ND	0.012	0.028	0.055	0.015
17	2144	** Histidine **	0.384	0.425	0.827	0.203	ND	ND	0.287	0.122
18	2211	** Tryptophan **	ND	0.148	0.076	ND	0.068	0.046	ND	0.044
		**Total**	**4.729**	**6.117**	**8.825**	**2.607**	**0.283**	**2.673**	**2.851**	**2.709**
**Organic acids**										
1	1305	Succinic acid	0.728	0.547	0.783	0.526	0.619	0.602	0.642	0.662
2	1344	Fumaric acid	0.548	0.412	0.589	0.396	0.466	0.453	0.483	0.498
3	1477	Malic acid	1.012	0.760	1.088	0.731	0.860	0.836	0.892	0.920
4	1818	Shikimic acid	1.662	1.249	1.787	1.201	1.413	1.374	1.465	1.511
5	1841	Citric acid	0.849	0.638	0.913	0.613	0.722	0.702	0.748	0.772
6	1855	Quinic acid	0.825	0.620	0.887	0.596	0.701	0.682	0.727	0.750
7	1946	L-Ascorbic acid	0.287	0.216	0.309	0.207	0.244	0.237	0.253	0.261
		**Total**	**5.911**	**4.441**	**6.357**	**4.271**	**5.025**	**4.885**	**5.211**	**5.374**
**Sugar alcohols**										
1	1932	Sorbitol	0.326	0.245	0.232	0.197	0.182	0.269	0.281	0.296
2	2034	Myo-inositol	0.187	0.141	0.104	0.127	0.091	0.187	0.157	0.170
**Saccharides (mono-, di-)**										
1	1856	Fructose isomer	0.972	1.355	1.588	1.984	1.804	1.491	1.389	1.640
	1865	Fructose isomer	0.337	0.470	0.550	0.688	0.625	0.517	0.481	0.568
2	1881	Glucose issomer	1.855	2.586	3.029	3.786	3.442	2.844	2.650	3.129
	1901	Glucose issomer	1.397	1.947	2.281	2.851	2.592	2.142	1.996	2.356
3	2620	Sucrose isomer (alpha-D-Glc-(1.2)-beta-D-Fru)	2.992	4.171	4.885	6.106	5.551	4.588	4.274	5.047
	2833	Sucrose isomer (alpha-D-Glc-(1.2)-beta-D-Fru)	1.737	2.421	2.836	3.545	3.223	2.663	2.481	2.930
		**Total**	**9.290**	**12.950**	**15.168**	**18.960**	**17.236**	**14.245**	**13.272**	**15.669**
**Saturated and unsaturated fatty acids**										
1	1719	Tetradecanoic acid (Myristic acid)	0.336	0.319	0.166	0.460	0.394	0.259	0.276	0.287
2	1926	n-Hexadecanoic acid (Palmitic acid)	3.953	3.754	1.948	5.411	4.625	3.040	3.246	3.378
3	2095	9,12-(Z,E)-Octadecadienoic acid (Linoleic acid)	2.451	2.327	1.208	3.355	2.867	1.885	2.013	2.095
4	2099	9-(Z)-Octadecenoic acid (Oleic acid)	1.285	1.220	0.633	1.758	1.503	0.988	1.055	1.098
5	2103	9,12,15-(Z,Z,Z)-Octadecatrienoic acid (Linolenic acid)	0.445	0.423	0.219	0.609	0.521	0.342	0.366	0.380
6	2247	n-Octadecanoic acid (Stearic acid)	1.648	1.565	0.812	2.256	1.928	1.268	1.354	1.409
7	2311	n-Eicosanoic acid (Arahydic acid)	0.756	0.718	0.372	1.034	0.884	0.581	0.621	0.646
8	2408	n-Docosanoic acid (Behenic acid)	0.939	0.891	0.463	1.285	1.098	0.722	0.771	0.802
		**Total**	**11.811**	**11.217**	**5.821**	**16.168**	**13.819**	**9.086**	**9.701**	**10.095**
**Phenolic acids**										
1	1836	Protocatechuic acid	0.073	0.187	0.033	0.107	0.042	0.123	0.136	0.273
2	1945	trans-p-Coumaric acid	0.157	0.142	0.122	0.096	ND	0.083	0.096	0.185
3	2103	trans-Ferulic acid	0.110	0.092	0.055	0.077	0.064	0.137	0.026	0.055
4	2140	trans-Caffeic acid	0.034	0.128	0.045	0.068	0.085	0.044	0.053	0.026
5	2254	trans-Sinapic acid	0.072	0.062	0.061	0.050	0.022	0.031	0.046	0.055
6	3191	Chlorogenic acid	2.816	3.559	1.638	0.842	2.502	2.150	1.214	1.363
		**Total**	**3.263**	**4.170**	**1.955**	**1.240**	**2.714**	**2.569**	**1.572**	**1.957**

ND—not detected in the sample, RI—retention index; G—“Gergana”, F—“Filina”, U—“Ufo 4”, JL—“July Lady”, L—“Laskava”, FQ—“Flat queen”, Evm—“Evmolpiya”, M—“Morsiani 90”. Essential amino acids are marked with blue color.

**Table 2 antioxidants-12-00205-t002:** Enzyme-inhibitory activities (α-glucosidase, lipase, α-amylase, and acetylcholinesterase (AChE)) of free and bound phenolics of eight peach varieties’ peel extracts, IC_50_, mg/mL.

Samples		α-Glucosidase	Lipase	α-Amylase	AChE
G	Free	-	-	-	-
	Acid hydrolysis	-	-	-	-
	Alkaline hydrolysis	3.4 ± 0.11 ^k^	-	26.67 *	26.67 *
F	Free	-	-	-	-
	Acid hydrolysis	-	-	31.51 *	-
	Alkaline hydrolysis	9.0 ± 0.13 ^h^	-	20.55 ± 0.51 ^a^	31.1 ± 0.22 ^a^
U	Free	39.7 ± 0.05 ^b^	-	-	-
	Acid hydrolysis	22.9 ± 0.14 ^d^	-	-	-
	Alkaline hydrolysis	5.9 ± 0.04 ^i^	-	15.17 *	15.17 *
JL	Free	-	-	-	-
	Acid hydrolysis	26.1 ± 0.54 ^c^	-	-	21.3 *
	Alkaline hydrolysis	5.1 ± 0.13 ^j^	-	17.38 ± 0.11 ^b^	17.8 ± 0.52 ^b^
L	Free	27.2 ± 0.11 ^c^	-	-	-
	Acid hydrolysis	14.1 ± 0.09 ^f^	-	31.51 *	-
	Alkaline hydrolysis	9.0 ± 0.10 ^h^	-	-	30.23 *
FQ	Free	49.3 ± 0.22 ^a^	-	-	-
	Acid hydrolysis	-	-	-	-
	Alkaline hydrolysis	3.1 ± 0.10 ^k^	-	-	-
Evm	Free	19.4 ± 0.12 ^e^	-	-	56.36 *
	Acid hydrolysis	-	-	-	-
	Alkaline hydrolysis	2.7 ± 0.08 ^l^	-	-	-
M	Free	11.6 ± 0.14 ^g^	-	-	55.67 *
	Acid hydrolysis	-	-	-	-
	Alkaline hydrolysis	2.6 ± 0.08 ^l^	-	-	-

“-” not detected; Different letters in the same column indicate statistically significant differences (*p* < 0.05), according to ANOVA (one-way) and the Tukey test (n = 3). *—a concentration that inhibits 30% of the corresponding enzyme under the described conditions. G—“Gergana”, F—“Filina”, U—“Ufo 4”, JL—“July Lady”, L—“Laskava”, FQ—“Flat queen”, Evm—“Evmolpiya”, M—“Morsiani 90”.

**Table 3 antioxidants-12-00205-t003:** Effect of free and bound phenolics of eight peach varieties’ peel extracts on antimicrobial potential toward bacteria, yeast and fungi.

Test Microorganism/ Samples	Sample (Inhibition Zone, mm) *																							
	G	F	U	JK	L	FQ	Evm	M	G	F	U	JK	L	FQ	Evm	M	G	F	U	JK	L	FQ	Evm	M
	Free								Acid Hydrolysis								Alkaline Hydrolysis							
Gram (+) bacteria																								
*Bacillus subtilis* ATCC 6633	-	-	-	8	-	-	8	8	9	10	10	10	10	10	10	9	10	11	9	9	10	10	9	10
*Staphylococcus aureus* ATCC 25923	-	-	-	-	-	-	-	-	-	-	-	-	-	-	-	-	8	9	8	8	-	-	-	-
*Listeria monocytogenes* NBIMCC 8632	-	-	-	-	-	-	-	-	8	8	8	8	8	9	9	8	8	9	8	9	9	-	-	-
*Enterococcus faecalis* ATCC 19433	-	-	-	-	-	-	-	-	-	-	-	-	-	-	-	-	-	11	-	8	-	-	-	-
Gram (−) bacteria																								
*Salmonella* *enteritidis* ATCC 13076	-	-	-	*-*	-	-	-	-	-	-	-	-	-	-	-	-	-	9	-	8	-	-	-	-
*Escherichia coli* ATCC 8739	-	-	-	-	-	-	-	-	-	-	-	-	-	-	-	-	-	-	-	-	-	-	-	-
*Proteus vulgaris* ATCC 6380	-	-	-	-	-	-	-	-	-	-	-	-	-	-	-	-	-	8	-	8	-	-	-	-
*Pseudomonas aeruginosa* ATCC 9027	11	11	10	12	12	9	9	-	12	12	12	15	15	9	9	-	13	14	13	13	13	8	-	9
Yeasts																								
*Candida albicans* NBIMCC 74	-	-	-	-	-	-	-	-	-	-	-	-	-	-	-	-	8	8	-	-	-	9	9	8
*Saccharomyces cerevisiae* ATCC 9763	-	-	-	-	-	-	-	-	-	-	-	-	-	-	8	8	8	8	-	-	-	8	8	8
Fungi																								
*Aspergillus niger* ATCC 1015	-	-	-	-	-	-	-	-	8	8	8	8	8	8	9	9	8	8	8	8	8	8	8	8
*Aspergillus flavus*	-	-	-	-	-	-	-	-	-	-	-	-	-	-	-	-	8	8	8	8	8	8	-	-
*Penicillium* sp.	-	-	-	-	-	-	-	-	9	9	8	9	9	8	8	8	9	9	9	9	9	-	8	-
*Rhizopus* sp.	10	10	-	10	10	-	-	-	8	8	-	10	10	-	10	10	9	9	-	-	-	-	8	-
*Fusarium moniliforme* ATCC 38932	-	-	-	-	-	-	-	-	8	8	8	9	9	8	8	8	10	10	10	10	10	-	-	-
*Mucor* sp.	-	-	-	-	-	-	-	-	-	-	-	-	-	-	-	-	-	-	-	-	-	-	-	-

* d_well_ = 6 mm; G—“Gergana”, F—“Filina”, U—“Ufo 4”, JL—“July Lady”, L—“Laskava”, FQ—“Flat queen”, Evm—“Evmolpiya”, M—“Morsiani 90”.

**Table 4 antioxidants-12-00205-t004:** Pearson correlation matrix of phenolic compounds and antioxidant activity of free and bound phenolics of eight peach varieties’ peel extracts.

Variables	TPC	TFC	TMA	DPPH	ABTS	FRAP	CUPRAC
TPC	1	0.3249 ***	0.5957 *	0.8856 *	0.5059 *	0.8231 *	0.6251 *
TFC		1	0.0608 ***	0.5701 *	0.1461 ***	0.7241 *	0.5234 *
TMA			1	0.6286 *	0.5618 *	0.4695 *	0.3833 *
DPPH				1	0.4303 *	0.9562 *	0.7453 *
ABTS					1	0.2979 **	0.2376 **
FRAP						1	0.7155 *
CUPRAC							1

TPC, total phenolics content; TFC, total flavonoids content; TMA, total monomeric anthocyanins; DPPH, antioxidant activity determined by the DPPH assay; ABTS, antioxidant activity determined by the ABTS assay; FRAP, ferric reducing antioxidant power; CUPRAC, cupric ion reducing antioxidant capacity. * Correlation is significant at *p* ≤ 0.01; ** correlation is significant at *p* ≤ 0.05; *** correlation is not significant (*p* > 0.05).

## Data Availability

The data presented in this study are available on request from the corresponding author.

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
