# Peer review of "Valorization of Peels of Eight Peach Varieties: GC–MS Profile, Free and Bound Phenolics and Corresponding Biological Activities"

_antioxidants, 2023, doi:10.3390/antiox12010205_

Round 1
Reviewer 1 Report
The article contains valuable information. The techniques used for the identification and quantification of the compounds are adequate and the results are consistent with them.
However, there is a detail in the approach of the study that would need, in my humble opinion, to be corrected. The authors have identified and quantified metabolites using a GC-MS. Throughout the article they comment that they have quantified compounds and provide concentration data for each of them. However, the authors have used an internal standard to obtain these data (rititol and nonadecanoic acid, for each of the phases, polar and non-polar, respectively).
Authors should take into account that the use of 1 single standard cannot be valid to provide substance quantification data. Each of the compounds reacts differently to concentration variations with respect to the equipment signal, so the factors are not the same.
The authors should correct in their study the way in which they refer to the results. Perhaps the most accurate thing to do would be to refer to them as semi-quantification instead of quantization. In order to issue results referring to concentration, calibration lines should have been made for each of the compounds in order to inhibit the effect of the chemical family on the concentration factor.
This should be clear throughout the entire article.
In addition, there are some small considerations that I think should be taken into account:
- Table 1. There are some compounds in blue color. The reason for this color should be explained in the table footer or the color should be removed.
- Figure 1. The authors explain in the results that there are significant differences between the samples and the different extracts. However, the figure did not reflect the statistical treatment used. Statistical differences should be included in the graphs to make them self explanatory.
- Figure 2. Statistical data (groupings using letters, according to Tukey's test) are provided but deviations from the sample values suggest that the groupings are not valid. I explain. For example, figure 2B, the sample L and U, the Free value presents significant differences between both samples but the deviations are so large that they should not suggest these differences.
- The references section should be reviewed. There are years in bold and others without it, there are complete doi and others reduced...
Author Response
The authors would like to thank the reviewer for the critical notes and the helpful recommendations. We accept the suggestions and corrections. The text of the manuscript is now corrected according to the suggestions and thus we hope to implement the highly respectable recommendations.
With the corrections made we hope our paper to be found as suitable for publication.
Our detailed answers to the critics and questions are as follows:
- The article contains valuable information. The techniques used for the identification and quantification of the compounds are adequate and the results are consistent with them. However, there is a detail in the approach of the study that would need, in my humble opinion, to be corrected. The authors have identified and quantified metabolites using a GC-MS. Throughout the article they comment that they have quantified compounds and provide concentration data for each of them. However, the authors have used an internal standard to obtain these data (rititol and nonadecanoic acid, for each of the phases, polar and non-polar, respectively). Authors should take into account that the use of 1 single standard cannot be valid to provide substance quantification data. Each of the compounds reacts differently to concentration variations with respect to the equipment signal, so the factors are not the same.
Answer: The authors would like to thank the reviewer for his/her adequate constatation. Mass Spectrometry-based plant metabolomics experiments deal with two main approaches for quantitative analyses, namely, relative and absolute quantification. Relative quantification can be defined as the analyte instrumental response relative to an internal standard. This approach includes (1) the added amount of the exogenous internal standard compound (for example unlabelled ribitol (as in our case) or isotopically labelled 13C-sorbitol); (2) a description of the method used to evaluate instrument response (e.g. peak integration and deconvolution method - performed through AMDIS in our case); (3) the number of replicate analyses (five independent repetitions) and standard error. Also the determination of the relative response ratio is calculated using the metabolite peak area divided by both the peak area of the internal standard and the sample fresh/dry weight. We absolutely support that each of the metabolites in the samples reacts differently to concentration variations with respect to the equipment signal, so the factors are not the same. Following the next recommendation appropriate changes have been introduced in terms of results.
- The authors should correct in their study the way in which they refer to the results. Perhaps the most accurate thing to do would be to refer to them as semi-quantification instead of quantization. In order to issue results referring to concentration, calibration lines should have been made for each of the compounds in order to inhibit the effect of the chemical family on the concentration factor. This should be clear throughout the entire article.
Answer: The necessary changes and clarifications have been introduced throughout the manuscript.
- Table 1. There are some compounds in blue color. The reason for this color should be explained in the table footer or the color should be removed.
Answer: An explanatory footer has been added to the table.
- Figure 1. The authors explain in the results that there are significant differences between the samples and the different extracts. However, the figure did not reflect the statistical treatment used. Statistical differences should be included in the graphs to make them self explanatory.
Answer: The authors initially thought that the figure was presentable and informative but we have made changes to Figure 1 so that it can be presented more clearly as the reviewer has suggested.
- Figure 2. Statistical data (groupings using letters, according to Tukey's test) are provided but deviations from the sample values suggest that the groupings are not valid. I explain. For example, figure 2B, the sample L and U, the Free value presents significant differences between both samples but the deviations are so large that they should not suggest these differences.
Answer: Due to a technical error (fixed SD applied by program, instead of calculated ones) the data had to be corrected. We would like to thank the reviewer for the attentive review so that this mistake could be adequately addressed.
- The references section should be reviewed. There are years in bold and others without it, there are complete doi and others reduced...
Answer: The reference section has been revised as the reviewer has suggested.
Reviewer 2 Report
This is a very good and interesting paper.
Only two points should be clarified:
Line 146 "These extraction steps were repeated twice more". Please, clarify why the authors repeated the process.
Did the authors optimise the extraction process of free phenolic compounds before?
Author Response
The authors would like to show their appreciation for the reviewer’s helpful, attentive, and thorough review. We are delighted that we have provided an interesting scientific topic. We accept his\her very detailed suggestions and improvements for the manuscript.
Below, the reviewer can find a point by point answer to all comments and questions posed:
- Line 146 "These extraction steps were repeated twice more". Please, clarify why the authors repeated the process.
Answer: The authors would like to thank the reviewer for posing this question. A triple extraction is a common practice for extraction (and purification) in order to achieve better yield of target components. Some clarifications were added to the text.
- Did the authors optimise the extraction process of free phenolic compounds before?
Answer: We have not previously optimized the extraction process. We see the results from this study as a stepping stone for future optimization, which will be important for the valorization of fruit peels as beneficial phytochemicals.
We very much hope that the improved manuscript will be accepted for publication.
Round 2
Reviewer 1 Report
the authors have met all the demands suggested in the first review. The article has been improved and the pertinent clarifications have been added so that the reader understands the techniques used and the results obtained.
In my opinion, it can be accepted for publication.